

# Password authenticated key exchange-based on Kyber for mobile devices

Kübra Seyhan[1], Sedat Akleylek[2,3] and Ahmet Faruk Dursun[1]

[1] Department of Computer Engineering, Ondokuz Mayis University Samsun, Samsun, Turkey
[2] Chair of Security and Theoretical Computer Science, University of Tartu, Tartu, Estonia
[3] Department KEM of Computer Engineering, Istinye University, Istanbul, Turkey

## ABSTRACT

In this article, a password-authenticated key exchange (PAKE) version of the National Institute of Standards and Technology (NIST) post-quantum cryptography (PQC) public-key encryption and key-establishment standard is constructed. We mainly focused on how the PAKE version of PQC standard Kyber with mobile compatibility can be obtained by using simple structured password components. In the design process, the conventional password-based authenticated key exchange (PAK) approach is updated under the module learning with errors (MLWE) assumptions to add password-based authentication. Thanks to the following PAK model, the proposed Kyber.PAKE provides explicit authentication and perfect forward secrecy (PFS). The resistance analysis against the password dictionary attack of Kyber.PAKE is examined by using random oracle model (ROM) assumptions. In the security analysis, the cumulative distribution function (CDF) Zipf (CDF-Zipf) model is also followed to provide realistic security examinations. According to the implementation results, Kyber.PAKE presents better run-time than lattice-based PAKE schemes with similar features, even if it contains complex key encapsulation mechanism (KEM) components. The comparison results show that the proposed PAKE scheme will come to the fore for the future security of mobile environments and other areas.

# INTRODUCTION

The security of conventional public-key cryptosystems (PKC) changed with the post-quantum concept that emerged with ongoing processes for developing quantum computers and the proposal of the Shor algorithm. The traditional PKCs such as key exchange (KE)/KEM and digital signature schemes will be insecure in the presence of large-scale quantum computers with Shor algorithm (*Peikert, 2016*). NIST started a process to set the post-quantum secure standard for PKC in 2016 (*NIST, 2022a*). In 2022, lattice-based Kyber was determined as the standard in the KEM category. For digital signature usage, lattice-based Crystals-Dilithium, Falcon, and hash-based SPHINCS+ were selected as the standard (*NIST, 2022b*). Although the standards were determined to be ready PQC era, it is still necessary to design and determine cryptosystems that can be used for particular goals and application areas.

Corresponding author
Sedat Akleylek, akleylek@gmail.com

One of the PKC primitives used for specific purposes is the PAKE scheme that provide a high-entropy shared key generated using low-entropy password-based authentication. Due to the easy-to-use structure, PAKE schemes do not require special hardware to store high entropy keys (*Bellare, Pointcheval & Rogaway, 2000*). The hardness assumptions of these schemes are also based on discrete logarithm and factorization problems like other PKCs. The first PAKE, encrypted key exchange, was proposed by Bellovin and Merritt in 1992 (*Bellovin & Merritt, 1992*) and many PAKE proposals, including new theoretical models, were presented in the following years (*Bellovin & Merritt, 1993*; *Jablon, 1996*; *Wu, 1998*; *Hao & Ryan, 2011*; *Shin & Kobara, 2012*). In addition, Internet Engineering Task Force (IETF), The Institute of Electrical and Electronics Engineers (IEEE), and the International Organization for Standardization (ISO)/International Electrotechnical Commission (IEC) conducted studies on the standardization of PAKE protocols (*Hao & van Oorschot, 2022*). The most recent standardization initiative for PAKE schemes was the process initiated by the IETF in 2019. In this call, completed in March 2020, OPAQUE and CPace schemes were declared as the PAKE standard for today's usage (*Hao, 2021*). Although the industry has started to prototype PAKE protocols in real applications with these processes, the adaptation of post-quantum secure algorithms is necessary for future security.

With the development of wireless communication technologies, the increasing use of mobile devices has brought the security of these devices into focus. There is a need for post-quantum secure PKCs such as KEM, authenticated key exchange, and PAKE that consider resource limitations for mobile devices (*Dabra, Bala & Kumari, 2020*). Lattice-based cryptosystems stand out with their strong proof of security, worst-case hardness, efficiency, and post-quantum security features. Up-to-date literature shows that there have not been many lattice-based PAKEs for mobile device security. In *Dabra, Bala & Kumari (2020)*, an anonymous ring learning with errors (RLWE)-based two-party PAKE was designed for the post-quantum security of the mobile environment. The security analysis of this scheme, which includes a four-phase approach, was done by considering real-or-random (RoR) assumptions. An improved version of *Dabra, Bala & Kumari (2020)* with a practical randomized KE approach is proposed in *Ding, Cheng & Qin (2022)* to capture signal leakage attack resistance. In *Islam & Basu (2021)*, a four-phase RLWE-based PAKE was constructed for two mobile devices-one server communication model. The security-related examinations were done by following ROM definitions. In *Seyhan & Akleylek (2024)*, we also built a four-phase PAKE to achieve reusable key and anonymity features for mobile device-server communication model. In the security analysis, we followed RoR assumptions to prove the semantic security. According to the up-to-date studies, many other PAKEs with lattice primitives such as *Ding et al. (2017)*, *Gao et al. (2017)*, *Liu et al. (2019)*, *Seyhan & Akleylek (2023)* and *Ren, Gu & Wang (2023)* were designed using traditional PAK model to capture explicit authentication and PFS. The provided proposals can be suitable for post-quantum key agreement requirements, but none of them has been focused on the PAKE version of the NIST standard. We know that the security of Kyber has been deeply studied and it was designed with efficient structures. Therefore, proposing

a PAKE version of this algorithm and providing reference implementations will come to the fore in post-quantum secure PAKE literature.

## Motivation and contribution

PAKE protocols are commonly used for credential recovery, wireless fidelity communication, device pairing, end-to-end (E2E) secure channel applications, and Kerberos-like usage areas as a part of secure communication in daily life. It is known that ensuring today's and post-quantum security of PAKE schemes is one of the main open problem regarding security in the future (*Ott & Peikert, 2019*; *Hao & van Oorschot, 2022*). Although the strongest candidates can be built with NIST algorithms, PAKE versions of these schemes have not been constructed yet. To propose a solution for this open problem, we used well-defined Kyber KEM structures to construct password-based authentication. We mainly aimed to solve the post-quantum authenticated key-sharing requirement of traditional computing power and mobile devices by providing a PAKE version of the PQC standard Kyber scheme. The contributions of Kyber.PAKE proposal to the literature are listed as follows.

- A novel two-party Kyber.PAKE is constructed to meet the post-quantum secure PAKE requirement for general purposes and mobile networks based on NIST PQC KEM standard. The conventional PAK design suite (*MacKenzie, 2002*) is adapted to MLWE problem since the main security of Kyber is based on MLWE.
- KEM structures and MLWE-based PAK design idea are used simultaneously to construct the PAKE version of Kyber. So, the proposed Kyber.PAKE provides explicit authentication and PFS without using a trusted third party, public key infrastructure, and signature.
- The security of Kyber.PAKE is deeply analyzed by making some assumptions about whether an adversary can obtain the shared key with an online dictionary attack or not. In the analysis, the advantage of the adversary is shown to be negligible in the ROM by following the Bellare-Pointcheval-Rogaway (BPR) (*Bellare, Pointcheval & Rogaway, 2000*) and CDF-Zip models (*Wang et al., 2017*). Since CDF-Zipf characterizes password distribution, theoretical security analysis is performed by better covering the real-world power of the adversary.
- The implementation of the Kyber.PAKE is written in C (*Dursun, 2023a*) and Java (*Dursun, 2023b*). The experimental results are presented in terms of cost, central process unit (CPU) cycle, and run-time. Based on Java implementation, the mobile device performance are also provided by considering running time, energy, memory, and CPU usages.
- Reference results show that the proposed Kyber.PAKE is one of the best choices to meet authenticated key generation requirement of post-quantum era with the usage of simple structure PAKE design and KEM with strong security.

## Outline

In 'Preliminaries', the mathematical background is summarized. In 'Proposed Kyber.PAKE Scheme', the general working steps and correctness of the constructed Kyber.PAKE are

**Table 1  Notations.**

| | |
|---|---|
| $\mathbb{Z}_q$: Integers in modulo $q$. | $R^k$: k-dimensional vector of polynomials ($R$). |
| $\mathrm{mod}^+$: Let $\alpha \in \mathbb{Z}^+$. $a' = a \bmod {}^+\alpha \| a' \in [0,\ldots,\alpha)$. | $R_q^k$: $R^k$ in mod $q$ |
| $\|$: Concatenation operator. | $\kappa$: Security parameter. |
| $B^\ell$ - $B^*$: Byte array of length $\ell$ and arbitrary, respectively. | $D_{k,\eta}^{\mathrm{MLWE}}$: MLWE distribution. |
| $\psi_{d \in \{d_t,d_v,d_u\}}^k$: The correctness distribution of Kyber. | $B_\eta$: CBD of Kyber. Let $\eta \in \mathbb{Z}^+$. For $\{(a_i,b_i)\}_{i=1}^\eta \leftarrow (\{0,1\}^2)^\eta$, a $B_\eta$ sample is obtained with $\sum_{i=1}^\eta (a_i - b_i)$. |
| $b_\eta^k$: $B_\eta$ distribution over $R^k$. | $d_t, d_v, d_u$: Reconciliation parameters of Kyber. |
| $pw_C$: Client's password. | $a \leftarrow^r \chi$: $a$ is randomly chosen from the distribution $\chi$. |
| sid - cid: Server id - Client id. $C$ - $S$ - $V$: Client - Server - Participant Spaces. | $H_1(\cdot) = \mathrm{SHAKE} - 128 : \{0,1\}^* \to R_q^k$. |
| $\epsilon$: A negligible value in $\kappa$. | $H_2(\cdot) = \mathrm{SHA3} - 256 : \{0,1\}^* \to \{0,1\}^k$. |
| $U(\cdot)$: Uniform distribution. | $\mathrm{mod}^\pm$: Modular reduction. Let $\alpha \in 2\mathbb{Z}^+.a' = a \bmod {}^\pm\alpha \| a' \in (-\alpha/2,\ldots,\alpha/2]$. |
| $H_3(\cdot) = \mathrm{SHA3} - 256 : \{0,1\}^* \to \{0,1\}^k$ Key derivation function (KDF) is used to obtain $k$-bit session key. | pk - sk: Public key - Secret key. |
| | negl($\kappa$): Let $\varpi > 0$ and $\kappa > n_0$. If an $n_0 \in \mathbb{N}$ can be found such that negl($\kappa$) $< \kappa^{-\varpi}$, negl is determined as a negligible function. |
| $D_{pk}$: $pk$ distribution of Kyber KEM defined with $B^{12kn/8+32}$. | $D_{ct}$: $ct$ distribution of Kyber KEM defined with $B^{d_u kn/8+d_v n/8}$. |
| CCA: Chosen-ciphertext attack. | XOF: Extendable Output Function |
| $NTT$: Number-Theoretic Transform. | CPA: Chosen-plaintext attack. |
| $NTT^{-1}$: Inverse NTT. | PKE: Public Key Encryption. |
| PFR: Pseudo-random function. | Adv: Advantage |
| **A**: Adversary | CBD: Centered Binomial Distribution. |
| ssk - ct: Shared secret key - Ciphertext. | **S**: Abbreviation of Kyber.PAKE. |

defined. In 'Security Analysis', the detailed security examinations against dictionary attacks is presented. The implementation results and comparison with current literature are provided in 'Reference Implementation and Comparison Results'. In the last part, 'Conclusion and Future Directions', the future directions and conclusion are figured out.

## PRELIMINARIES

The notation is provided in Table 1.

### Basic definitions

In the proposed PAKE, the shared key is obtained by using Kyber PKE and KEM functions/components and the password-based authentication is added by following PAK design idea.

Kyber PKE and KEM functions are recalled in Table 2. To obtain detailed information, we refer to *Avanzi et al. (2019)*.

In Table 2, KYBER.CCAKEM uses KYBER.CPAPKE functions to obtain key agreements based on the MLWE problem. Since the main security of Kyber and the proposed PAKE

**Table 2 Kyber KEM and PKE structures.** (*Avanzi et al., 2019*).

| KYBER.CCAKEM.KeyGEN() | KYBER.CCAKEM.Enc(pk) | KYBER.CCAKEM.Dec(c, sk) |
|---|---|---|
| **Output:** sk$\in$B$^{24kn/8+96}$ | **Input:** pk$\in$B$^{12kn/8+32}$ | **Input:** $c \in$B$^{d_ukn/8+d_vn/8}$, sk$\in$B$^{24kn/8+96}$ |
| **Output:** pk$\in$B$^{12kn/8+32}$ | **Output:** $c \in$B$^{d_ukn/8+d_vn/8}$, $K \in$B$^*$, where K is ssk | **Output:** $K \in$B$^*$, where K is ssk. |
| $z \leftarrow$B$^{32}$ | $m \leftarrow$B$^{32}$ | pk = sk $+ 12 \cdot k \cdot n/8$ |
| (pk,sk$'$) =KYBER.CPA.PKE.KeyGEN() | $m \leftarrow H(m)$ | h = sk $+ 24 \cdot k \cdot n/8 + 32$ |
| sk = (sk$'$||pk||H(pk)||z) | $(\bar{K},r) = G(m||H(pk))$ | z = sk $+ 24 \cdot k \cdot n/8 + 64$ |
| **return** (pk,sk) | $c$ =KYBER.CPAPKE.Enc(pk,$m,r$) | $m'$ = KYBER.CPAPKE.Dec($\mathbf{s}$, $(\mathbf{u},v)$) |
| | $K = $ KDF($\bar{K}$||$H(c)$) | $(\bar{K}',r') = G(m'||h)$ |
| **KYBER.CPAPKE.KeyGEN()** | **return**$(c,K)$ | $c'$ = KYBER.CPAPKE.Enc(pk,$m',r'$) |
| **Output:** sk$\in$B$^{12kn/8}$ | | **if** $c = c'$ **then return** $K = $KDF($\bar{K}'$||$H(c)$) |
| **Output:** pk$\in$B$^{12kn/8+32}$ | **KYBER.CPAPKE.Enc(pk,$m,r$)** | **else return** $K = $KDF($z$||$H(c)$) |
| $d \leftarrow$B$^{32}$ | **Input:** pk$\in$B$^{12kn/8+32}$, $m \in B^{32}$, $r \in$B$^{32}$ | **return** $K$ |
| $(\rho,\sigma) = G(d)$ | **Output:** $c \in$B$^{d_ukn/8+d_vn/8}$ | |
| $\hat{A} \in R_q^{k \times k}$ | $\hat{t} = $ Decode$_{12}$(pk) | **KYBER.CPAPKE.Dec(sk,$c$)** |
| $s,e \in R_q^k($B$_{\eta_1})$ | $\rho = $ pk $+ 12 \cdot k \cdot n/8$ | **Input:** $c \in$B$^{d_ukn/8+d_vn/8}$, sk$\in$B$^{12kn/8}$ |
| $\hat{s} = NTT(s)$, $\hat{e} = NTT(e)$ | $\hat{A} \in R_q^{k \times k}$ | **Output:** $m \in B^{32}$ |
| $\hat{t} = \hat{A} \circ \hat{s} + \hat{e}$ | $r \in R_q^k($B$_{\eta_1})$ | $\mathbf{u} = $ Decompress$_q$(Decode$_{d_u}(c),d_u$) |
| pk = (Encode$_{12}(\hat{t}$ mod $^+q$)||$\rho$) | $e_1 \in R_q^k($B$_{\eta_2})$ | v = Decompress$_q$(Decode$_{d_v}(c + d_u \cdot k \cdot n/8),d_v$) |
| sk = (Encode$_{12}(\hat{s}$ mod $^+q$)) | $e_2 \in R_q($B$_{\eta_2})$ | $\hat{s} = $ Decode$_{12}$(sk) |
| **return** (pk,sk) | $\hat{r} = $ NTT($r$) | $m = $ Encode$_1$(Compress$_q$($v - NTT^{-1}(\hat{s}^T \circ $NTT($u$)),1)) |
| | $\mathbf{u} = NTT^{-1}(\hat{A}^T \circ \hat{r}) + e_1$ | **return** $m$ |
| | $v = NTT^{-1}(\hat{t}^T \circ \hat{r}) + e_2 + $Decompress$_q$(Decode$_1(m)$,1) | |
| | $c_1 = $ Encode$_{d_u}$(Compress$_q(\mathbf{u},d_u)$) | • XOF is used in key generation |
| | $c_2 = $ Encode$_{d_v}$(Compress$_q(v,d_v)$) | • $m$: Message, $c$: Ciphertext |
| | **return** $c = (c_1||c_2)$ | • $H$ :B$^* \rightarrow$B$^{32}$ and $G$ :B$^{32} \rightarrow$B$^{32}$ |

version are based on the hardnesses of MLWE, the key generation is done by following the MLWE assumption.

**Definition 1 (MLWE (*Bos et al., 2018*))** *Let $k \in \mathbb{Z}^+$, $a_i \leftarrow^r R_q^k$, $s \leftarrow^r b_\eta^k$, and $e_i \leftarrow^r b_\eta$. MLWE distribution is obtained as follow. $D_{k,\eta}^{\mathrm{MLWE}} : (a_i, b_i = a_i^T s + e_i) \in R_q^k \times R_q$*

The hardness of MLWE is defined by decisional-MLWE (d-MLWE). Let $m$ independent $(a_i, b_i)$ instances are given $(A \in R_q^{m \times k}, b \in R_q^m)$. d-MLWE is a problem that decides whether these samples belong to MLWE ($D_{m,k,\eta}^{\mathrm{MLWE}} : (A, b = As + e)$, where $s \leftarrow^r b_\eta^k$ and $e_i \leftarrow^r b_\eta^m$) or uniform distribution ($U(R_q^{m \times k}) \times U(R_q^m)$).

Let $\mathbf{A}$ be an adversary. The advantage (Adv) of $\mathbf{A}$ to solve d-MLWE problem is determined by

$$\mathrm{Adv}_{m,k,\eta}^{\mathrm{MLWE}}(\mathbf{A}) = \left| \Pr[b' = 1 : b' \leftarrow \mathbf{A}((A,b) \in D_{m,k,\eta}^{\mathrm{MLWE}})] - \right.$$

$$\left. \Pr[b' = 1 : b' \leftarrow \mathbf{A}((A,b) \in U(R_q^{m \times k}) \times U(R_q^m))] \right|$$

In Table 2, the computations of pk and ct are done by discarding low-order bits that don't affect the accuracy of decryption to achieve reconciliation and reduced parameters. The reconciliation functions of Kyber are recalled in Definition 2 (*Bos et al., 2018*).

**Definition 2 (Compress and Decompress Functions (*Bos et al., 2018*))** *Let $a \in \mathbb{Z}_q$ and $d < \lceil \log_2(q) \rceil$.*

- $b = $*Compress* $_q(a,d)$*: For $a \in \mathbb{Z}_q$, the output of Compress is defined by $b = \lceil \frac{2^d}{q} \cdot a \rfloor$ mod $^+2^d$.*
- $b' = $*Decompress* $_q(b,d)$*: For $b \in \{0, \ldots, 2^d - 1\}$, the output of Decompress is determined by $b' = \lceil \frac{q}{2^d} \cdot b \rfloor$, where $b'$ is an element which is relatively close to $b$.*

The distribution $|b' - b \mod {}^{\pm}q| \leq B_q = \lceil q/(2^{d+1}) \rceil$ is nearly uniform over the integers of maximum magnitude $B_q$. Note that Definition 2 is defined over $\mathbb{Z}_q$. In Kyber, since $a \in R_q^k$, for each coefficient of $a$ is evaluated under these functions.

**Remark 1** *In Kyber (Bos et al., 2018), the reconciliation is provided by using the Compress and Decompress functions. So, $\psi_d^k$ is defined to satisfy the correctness. The output of distribution $\psi_d^k$ is generated in the following way.*

  i. *A $y \leftarrow^r R^k$ is chosen.*
 ii. ***return** $(y - Decompress_q((Compress_q(y,d)),d)) \mod {}^{\pm}q$.*

Although the main operations of Kyber are performed in the NTT domain, all polynomials are sent in the normal domain. For the transformation of polynomials to be used in the protocol flow, encode and decode operations are done (*Bos et al., 2018*; *Avanzi et al., 2019*).

**Definition 3 (*Decode$_\ell$*):** *Let $B^{32\ell}$ be a byte array. Then the output of Decode$_\ell$ is defined by $f = f_0 + f_1 X + f_2 X^2 + \cdots + f_{255} X^{255}$, where $f_i \in \{0, \ldots, 2^\ell - 1\}$. In other words, it deserializes a $32\ell$ bytes array into a polynomial with $B^{32\ell} \to R_q$.*

Note that Encode$_\ell$ is determined as the reverse of Decode$_\ell$.

The correctness of Kyber.PAKE is analyzed by using the correctness assumptions of KYBER.CCAKEM and KYBER.CPAPKE. The main theorems of these schemes are recalled in Theorems 1 and 2, respectively.

**Theorem 1** *Let $k \in \mathbb{Z}^+$, $\{s, e, r, e_1\} \leftarrow b_\eta^k$, $e_2 \leftarrow b_\eta$, $c_t \leftarrow \psi_{d_t}^k$, $c_u \leftarrow \psi_{d_u}^k$, $c_v \leftarrow \psi_{d_v}$, and $\delta = Pr[||e^T r + c_t^T r - s^T e_1 - s^T c_u + e_2 + c_v||_\infty \geq \lceil q/4 \rceil]$. Then, KYBER.CPAPKE scheme runs with $(1 - \delta)$ correctness probability (Bos et al., 2018).*

**Theorem 2** *Let $G$ be a random oracle (RO) and KYBER.CPAPKE is correct with $(1 - \delta)$ probability. KYBER.CCAKEM also runs with $(1 - \delta)$ correctness probability (Bos et al., 2018).*

The security evaluations of Kyber.PAKE is presented based on the ROM assumptions of Kyber.

**Definition 4 (ROM Security of Kyber KEM (*Avanzi et al., 2019*))** *Let XOF, H, and G be the ROs, $n_{ro}$ be the maximum number of **A**'s queries to ROs, and B–C be the adversaries who have roughly the same run-time as **A**. The adventage(Adv) of **A** over Kyber KEM in the ROM is defined by Eq. (1)*

$$\text{Adv}_{\text{KyberKEM}}^{\text{CCA}}(\mathbf{A}) = 2\text{Adv}_{k+1,k,\eta}^{\text{MLWE}}(B) + \text{Adv}_{\text{PRF}}^{\text{prf}}(C) + 4n_{ro}\delta \tag{1}$$

## Security model

In this section, special terms and basic primitives of the used security model are detailed.

In the construction of Kyber.PAKE, password-related primitives are added to provide main authentication by adapting traditional PAK (*MacKenzie, 2002*) design to the MLWE problem. In the analysis, the resistance against password dictionary attacks is investigated with the help of BPR (*Bellare, Pointcheval & Rogaway, 2000*) definitions.

- $C \in C, S \in S, V \in V = C \cup S$.
- *DS* denotes password space which is constructed according to Zipf's rule (*Wang et al., 2017*).
- Each $C$ has $pw_C \leftarrow^r DS$ and related $S$ holds the hash of $pw_C$.
- **A** is designed as a probabilistic algorithm, which can control the entire network and provide input for the participant's instances.
- By using the RO queries, **A** can launch the attacks.
- Let **S** be a scheme and $\prod_V^i$ be $i$th $V$ instance that can only be used once. **A**'s special query band is defined as follows.

  - execute($C, i, S, j$): **S** occurs between $\prod_C^i$ and $\prod_S^j$. The outputs of executed **S** are sent to **A**.
  - send($V, i, M$): Message $M$ is sent to $\prod_V^i$. Then, according to **S**, the computations of the scheme are done by $\prod_V^i$. The outputs are sent to **A**.
  - reveal($V, i$): Let $\prod_V^i$ be an accepted and has its own ssk. As a result of this query, ssk is sent to **A**.
  - corrupt($V$): It returns the password of $V$. If $V \in C$, the output is $pw_C$. Otherwise, $H_1(pw_C)$.
  - test($V, i$): Let $b$ be the coin of $\prod_V^i$. With this query, **A** tosses $b$. If $b = 0$, ssk is sent to **A** by $\prod_V^i$. Otherwise, ssk is chosen uniformly at random from ssk space and is returned to **A**.

- p-id and s-id are the id's of the parties and a session, respectively.
- $n_e$, $n_s$, $n_r$, $n_c$, and $n_o$ represent the maximum number of **A**'s execute, send, reveal, corrupt, and RO queries, respectively.
- $T_{\exp}$ represents the generation time of the MLWE samples.

According to the BPR model, each user can run the scheme multiple times with different partners.

**Definition 5 (Instance Partnership (*Bellare, Pointcheval & Rogaway, 2000*))** *Let $\prod_U^i$ and $\prod_V^j$ have(p-id$_i$, s-id$_i$, ssk$_i$) and (p-id$_j$, s-id$_j$, ssk$_j$), respectively. If the following conditions are satisfied $\prod_U^i$ and $\prod_V^j$ are considered as partner instances.*

- $U \in C$ and $V \in S$, or $V \in C$ and $U \in S$.
- $ssk_i = ssk_j$, p-id$_i = V$, and p-id$_j = U$.
- s-id$_i =$ s-id$_j =$ s-id, where this value is not null.
- A third oracle other than $\prod_U^i$ and $\prod_V^j$ should not have the same s-id.

In the security analysis, the instance freshness provides PFS.

**Definition 6 (Instance Freshness (*Bellare, Pointcheval & Rogaway, 2000*; *MacKenzie, 2002*))** *Let $\prod_W^i$ and $\prod_V^j$ be partner. If none of the following events occurred, $\prod_W^i$ is defined as a fresh instance that provide forward secrecy.*

- *A* reveal$(W,i)$ *query*
- *A* reveal$(V,j)$ *query*
- *A* corrupt$(V)$ *query before* send$(W,i,M)$ *and* test$(W,i)$ *queries.*

By using definitions and query band, the advantage of **A** in the PAKE scheme is examined.

**Definition 7 (Advantage of an A (*Bellare, Pointcheval & Rogaway, 2000*; *MacKenzie, 2002*))** *Let $\prod_V^i$ be a fresh instance, **S** be the PAKE scheme, and* $\mathrm{Suc}_{\mathrm{PAKE}}^{\mathbf{S}}$ *be an event that* **A** *makes a* $b' = \mathrm{test}(V,i)$ *query. For* $b$ *that was selected in the test query, if* $b' = b$, *the advantage of* **A** *is defined by* Eq. (2)

$$\mathrm{Adv}_{\mathrm{PAKE}}^{\mathbf{S}}(\mathbf{A}) = |2\mathrm{Pr}[\mathrm{Suc}_{\mathrm{PAKE}}^{\mathbf{S}}] - 1| \tag{2}$$

If the security analysis show that Eq. (2) is negligible, then the constructed PAKE is said to be secure under the ROM assumptions.

In the traditional PAK suit, the main advantage of the adversary is determined by considering that the password and uniform distribution have the same properties. Since this idea does not cover the real power of the adversary, CDF-Zipf is used to characterize the password distribution.

**Definition 8 (CDF-Zipf Model (*Wang et al., 2017*))** *Let DS be the password dictionary size and $n_{\mathrm{op}}$ be the maximum number of **A**'s online password guess attempts. In the traditional approach, the propability of **A**'s correct password guess is defined by $\frac{n_{\mathrm{op}}}{DS} + \mathrm{negl}(\kappa)$. According to the recent studies (*Wang et al., 2017*), this evaluation underestimate **A**'s power in real-world applications since the passwords of users generally follows CDF distribution. So, CDF-Zipf is followed to give more real-world-based results in terms of password distribution.*

*Let $C'$ and $f$ be CDF constants. The probability of **A**'s correct password guess in CDF-Zipf model is determined by*

$$Pr[\mathrm{Correctpw}] = C' \cdot n_{\mathrm{op}}^f + \mathrm{negl}(\kappa), \text{ where } C' \in [0.001, 0.1] \text{ and } f \in [0.15, 0.30] \tag{3}$$

*Note that CDF constants are determined according to the usage area by using linear regression.*

## PROPOSED KYBER.PAKE SCHEME

The password-authenticated version of Kyber KEM (*Avanzi et al., 2019*) is obtained with the combination of KYBER.CCAKEM.KeyGen, KYBER.CCAKEM.Enc, and KYBER.CCAKEM.Dec structures, given in Table 2, and MLWE-based one-phase PAK idea. The proposed Kyber.PAKE runs between client ($C$) and server ($S$) and contains four main sub-processes ($C_0$, $S_0$, $C_1$, $S_1$). The constructed scheme is detailed in Fig. 1. Let's clarify the design step of the proposed Kyber.PAKE for each sub-processes.

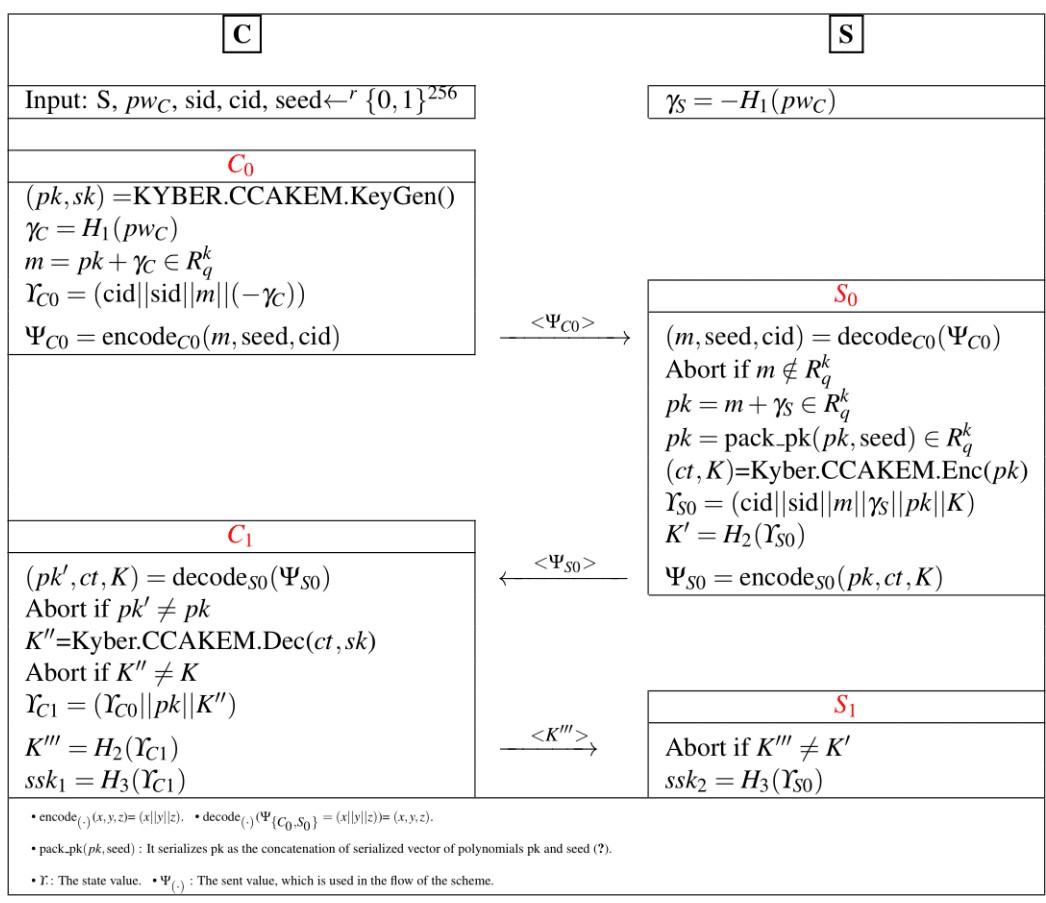

**Figure 1** Proposed Kyber.PAKE Scheme.

- **Phase $C_0$:** The key pairs $(pk, sk)$ are computed according to Kyber's MLWE-based key generation procedures with the help of KYBER.CCAKEM.KeyGen() and KYBER.CPAPKE.KeyGen() functions, defined in Table 2. After the computation of raw $pk$, the client generates and sends the encapsulated pk $(m = pk + \gamma_C)$.
- **Phase $S_0$:** On the server side, there is no public key computation like client side and the server retrieves raw pk $(pk = m + \gamma_S)$ using the password-related term. The key component of the server $(K)$ is determined with the usage of the encapsulation procedure of Kyber. The server computes $(ct, K) =$ Kyber.CCAKEM.Enc$(pk)$ and sends $K$ to provide authentication check in the client side.
- **Phase $C_1$:** The client retrieves sent values by using decode function and solves the $K''$ with help of Kyber's decapsulation $K'' =$ Kyber.CCAKEM.Dec$(ct, sk)$, where $K$ is equal to $K''$. By making authentication checks, the final password-authenticated shared key $ssk_1 = H_3(\overbrace{(\text{cid}||\text{sid}||m||(-\gamma_C))}^{\Upsilon_{C0}}||pk||K'')$ is generated.
- **Phase $S_1$:** The server makes comparision to ensure the authentication and generates $ssk_2 = H_3(\text{cid}||\text{sid}||m||\gamma_S||pk||K)^{\Upsilon_{S0}}$.

In the proposed PAKE, Compress, and Decompress functions, defined in Definition 2, are used to solve the reconciliation problem as a part of Kyber.CCAKEM.Enc and Kyber.CCAKEM.Dec procedures and $K = K''$ equality is obtained.

Let's deeply analyze the relationship between these two terms to show which conditions the proposed scheme will run correctly.

- In Fig. 1, if $K = K''$ is satisfied for $(ct, K) =$Kyber.CCAKEM.Enc$(pk)$ and $K''$ =Kyber.CCAKEM.Dec$(ct, sk)$, the correctness of Kyber.PAKE is also captured.
- In the Kyber.PAKE, $pk$ is retrieved by using the password component. In the $S_0$ phase, if $pk = m + \gamma_S$ is correctly solved with the help of $m$, there is no changes on the correctness of Kyber.
- Let's prove the correctness of Kyber.PAKE based on Theorems 1 and 2.

**Claim 1** *Let Kyber KEM be correct with $(1 - \delta)$ probability (Bos et al., 2018). Then, Kyber.PAKE scheme will also run correctly with $(1 - \delta)$ probability.*

**Proof 1** According to the detailed definition of and Kyber.CCAKEM.Enc in *Bos et al. (2018)*, it uses Kyber.CPAPKE.Enc procedure to generate $(ct, K)$, where $ct = (u, v)$. In Fig. 1, the input of Kyber.CCAKEM.Enc is $pk$ and computed with $pk = m + \gamma_S$. Since if the server correctly recover the $m$ from $pk$ with $pk = m + \gamma_S = pk + \gamma_C + \gamma_S$, where $\gamma_C = -\gamma_S$. By rewriting Remark 1 in *Bos et al. (2018)*, Eq. (4) is obtained.

$$t = \text{Decompress}_q(\text{Compress}_q(\overbrace{m + \gamma_S}^{pk + \gamma_C}, d_t), d_t) = As + e + c_t$$
$$u = \text{Decompress}_q(\text{Compress}_q(A^T r + e_1, d_u), d_u) = A^T r + e_1 + c_u$$
$$v = \text{Decompress}_q(\text{Compress}_q(t^T r + e_2 + \lceil q/2 \rfloor \cdot M, d_v), d_v) \qquad (4)$$
$$= (\overbrace{t}^{As + e + c_t})^T r + e_2 + \lceil q/2 \rfloor \cdot M + c_v$$
$$= (As + e)^T r + e_2 + \lceil q/2 \rfloor \cdot M + c_v + c_t^T r,$$

where $c_t, c_u \in R^k, c_v \in R$

Since there is no component to change the idea of Remark 1 in *Bos et al. (2018)*, if

$$||\overbrace{e^T r + c_t^T r - s^T e_1 - s^T c_u + e_2 + c_v}^{\delta}||_\infty \geq \lceil \frac{q}{4} \rfloor,$$ then the correctness of Kyber.PAKE is satisfied with $(1 - \delta)$ probability.

## SECURITY ANALYSIS

In the security analysis, MLWE-based PAK components are used to show that **A**'s probability of obtaining information about the session key with an online dictionary attack is negligible. In the adapted security model, **A** can make the following client-action (CA) and server-action (SA) queries.

- $CA_0$: **A** does $CA_0$ action to instruct the unused $\prod_C^i$ instance to transfer the related components to $S$.
- $SA_1$: **A** does $SA_1$ action to transfer the messages to unused $\prod_S^j$ instance.

- $CA_1$: **A** does $CA_1$ action to transfer the related message to $\prod_C^i$ instance that waits the related components of the scheme.
- $SA_2$: **A** does $SA_2$ action to transfer the messages to unused $\prod_S^j$ instance that waits the final components of the scheme.

According to the MLWE-based PAKE security analysis, **A** can take on the role a $\prod_C^i$, a $\prod_S^j$, and partner $\prod_C^i - \prod_S^j$ instances by using the some actions and special events. In the examinations, we modified the password guess events regarding MLWE and Kyber structures and presented them in Table 3 as the constructed Kyber.PAKE relies on the hardness assumption of MLWE and uses the Kyber components.

The Kyber.PAKE's proof of security is conducted by showing that **A** is unable to obtain the new ssk with a non-negligible advantage than the online dictionary attack. The advantage of **A** is given in Theorem 3.

**Theorem 3** *Let the proposed Kyber.PAKE scheme in* Fig. 1 *be represented by S, the password dictionary's size be presented with DS,* $|R_q^k| = q^{nk}$, *and the running time of* **A** *be T. For* $T' = O(T + (n_o + n_s + n_e)T_{exp})$, *the advantage of* **A** *over the Kyber.PAKE scheme is given in* Eq. (5).

$$Adv_{Kyber.PAKE}^{S}(\mathbf{A}) \leq O\Big(\frac{(n_e + n_s)(n_e + n_s + n_o) + n_o}{q^{nk}} + \frac{n_s}{2^{\kappa}} + Adv_{KyberKEM}^{CCA}(\mathbf{A}) +$$

$$n_s Adv_{R_q^k}^{d-MLWE}(T', n_o)\Big) + C' \cdot n_{op}^f \tag{5}$$

**Proof 3** Following PAK security analysis (*MacKenzie, 2002*), schemes $\{S = S0, S1, \ldots, S6\}$ are used to prove Theorem 3. In each scheme, **A** gains a different feature to make an online dictionary attack. Finally, he/she can create a password guess in the **S6**. The security of the proposed scheme is examined by proving that the advantage of **A** obtaining the session key of a fresh instance will be smaller than an online dictionary attack.

**S0**: It is the original Kyber.PAKE scheme.

**S1**: Let $m$ or $pk$ be chosen randomly by honest participants. If these values already appeared in the previous schemes, **S1** halts and **A** fails.

Let $\epsilon_1 = \frac{O((n_e + n_s)(n_e + n_s + n_o))}{q^{nk}}$.

**Claim 2** *For any* **A**, $Adv_{Kyber.PAKE}^{S0}(\mathbf{A}) \leq Adv_{Kyber.PAKE}^{S1}(\mathbf{A}) + \epsilon_1$

**Proof 2** Let's define $E1$ and $E2$ to describe the random selection of $m$ and $pk$. For $E = E1 \bigvee E2$, if the event $E$ occurs, then **S1** is equal to **S0**.

- Let $E1$ be an event defined for $m = m_1 = m_2 = m_3 = m_4$ in the following cases.

  - By making $CA_0$ or execute, $m_1$ is obtained.
  - $m_2$ is generated by a previous $CA_0$ or execute.
  - $m_3$ is used as an input of previous $SA_1$.
  - $m_4$ is utilized in a previous query $H_{l \in \{2,3\}}(\cdot)$.

- Let $E2$ be an event determined for $pk = pk_1 = pk_2 = pk_3 = pk_4$ in the following cases.

  - By making $SA_1$ or execute, $pk_1$ is generated.

**Table 3  Special cases of security analysis.**

| Event | Input | Output |
|---|---|---|
| **Testpw(·)** | $<C,i,S,pw,1>$ | For some $\{m,pk,\gamma_S,ct,K\}$, **A** makes;<br>• An $H_l(C,S,m,\gamma_S,pk,K)$ query.<br>• $CA_0$ query for $\Pi_C^i$.<br>  ⋆ The output is $(m,\text{seed},\text{cid})$.<br>• $CA_1$ query for $\Pi_C^i$.<br>  ⋆ The input is $(pk,ct,K)$.<br>• An $H_1(pw)$ query. It returns $-\gamma_S$. |
| | $<S,j,C,pw,1>$ | For some $\{m,pk,\gamma_S,ct,K\}$, **A** makes;<br>• An $H_l(C,S,m,\gamma_S,pk,K)$ query.<br>• A premade $SA_1$ query is made for $\Pi_S^j$.<br>  ⋆ The input is $(m,\text{seed},\text{cid})$.<br>  ⋆ The output is $(pk,ct,K)$.<br>• An $H_1(pw)$ query. It returns $-\gamma_S$.<br>The associated value of this event is obtained with<br>• The output of $H_{l=\{2,3\}}(\cdot)$ for $\Pi_S^j$. |
| | $<C,i,S,j,pw>$ | For some $l \in \{2,3\}$;<br>Let $\Pi_C^i$ and $\Pi_S^j$ be mutually partner of each other.<br>• Testpw$(S,j,C,pw,l)$ and Testpw$(C,i,S,pw,l)$ events occur. |
| **Testpw!(·)** | $<C,i,S,pw>$ | For some $\{ct,K\}$,<br>• A $CA_1$ query occurs.<br>  ⋆ The input is $(pk,ct,K)$.<br>• As a result of $CA_1$, a Testpw$(C,i,S,pw_C,2)$ occurs. |
| | $<S,j,C,pw>$ | • A Testpw$(S,j,C,pw,3)$ event is occured, which is associated with $K'''$.<br>• By using $K'''$ as an input;<br>  ⋆ **A** makes $SA_2$ query for $\Pi_S^j$. |
| **Testpw*(·)** | $<S,j,C,pw>$ | For some $l \in \{2,3\}$,<br>• Testpw$(S,j,C,pw,l)$ event occurs. |
| **Testexecpw(·)** | $<C,i,S,j,pw>$ | Firstly, **A** makes<br>• An execute query which generates $m,pk,ct$.<br>• An $H_1(pw)$ query. It returns $-\gamma_S$.<br>Then, for $l \in \{2,3\}$, **A** makes<br>• An $H_l(C,S,m,\gamma_S,pk,K)$ query. |
| **Correctpw** | - | **A** makes a corrupt query after either of these two events occurs.<br>• Testpw!$(C,i,S,pw)$ event occurs for $\Pi_C^i$.<br>• Testpw*$(S,j,C,pw)$ event occurs. |
| **Correctpwexec** | - | For some $\{C,i,S,pw\}$,<br>• A Testexecpw$(C,i,S,j,pw)$ event occurs. |
| **Doublepwserver** | - | For some $\{S,j,C,pw \neq pw'\}$,<br>The following events occur before any corrupt query.<br>• Testpw*$(S,j,C,pw)$.<br>• Testpw*$(S,j,C,pw')$. |
| **Pairedpwguess** | - | For some $\{C,i,S,j\}$,<br>• A Testpw$(C,i,S,j,pw)$ occurs. |

- There is no any special input for associated event.

– $pk_2$ is obtained by a previous $SA_1$ or execute.

– $pk_3$ is utilized as an input of previous $CA_1$.

– $pk_4$ is used in a previous query $H_{l\in\{2,3\}}(\cdot)$.

Considering the events $E1$ and $E2$, it is necessary to examine whether $m$ and $pk$ are previously or newly generated. In these events, the actions CA $_0$ and SA $_1$ are related to send and $H_{l \in \{2,3\}}(\cdot)$ queries are associated with RO queries. The previously generated $m$ or $pk$ can be obtained by making send, execute, and RO queries. So, the probability of $m$ or $pk$ occurring in the previous session is $\frac{(n_e+n_s+n_o)}{|R_q^k|}$. Since new $m$ or $pk$ can be generated with send and execute, the maximum number of queries is $(n_e + n_s)$. Therefore, the probability that $E$ happens is $\epsilon_1 = \frac{O((n_e+n_s)(n_e+n_s+n_o))}{q^{nk}}$.

**S2**: Unlike **S1**, send and execute are replied without answering any RO queries. Afterward, if the RO query is made, the answers are generated as consistently as possible with send and execute. The possible queries and answers in **S2** are given in Algorithm 1.

Let $\epsilon_2 = \frac{O(n_s)}{2^{\kappa}} + \frac{O(n_o)}{|R_q^k|}$.

**Claim 3** *For any* **A** *,* $\mathrm{Adv}^{S1}_{\mathrm{Kyber.PAKE}}(\mathbf{A}) \leq \mathrm{Adv}^{S2}_{\mathrm{Kyber.PAKE}}(\mathbf{A}) + \epsilon_2$

**Proof 3** In **S2**, since $m$ and $pk$ are new due to **S1**, $H_{l \in \{2,3\}}(\cdot)$ is also new. Therefore, the main condition for distinguishing **S1** and **S2** is that **A** queries $H_l(\cdot)$ for $l \in \{2,3\}$. In Algorithm 1, there are two possible cases.

- Since **A** does not make any $H_1(pw_C)$, where $-\gamma_S = H_1(pw_C)$, the maximum number of $H_l(\cdot)$ queries **A** can make is $\frac{O(n_o)}{|R_q^k|}$.
- **A** makes $\mathrm{send}(C, i, K')$ or $\mathrm{send}(S, j, K''')$ queries using the actions $CA_0$, $CA_1$, $SA_1$, and $SA_2$ in Algorithm 1. Neither of these queries is the output of an $H_2(\cdot)$ query that would be a correct password guess. Therefore, the maximum probability that **A** can abort the samples is $\frac{O(n_s)}{2^{\kappa}}$.

So, Claim 3 is satisfied.

**S3**: Unlike **S2**, the consistency is not controlled against the query execute when an $H_{l \in \{2,3\}}$ is queried. In other words, the event $\mathrm{Textexecpw}(C, i, S, j, pw_C)$ is not checked. So, the scheme responds with a random output rather than maintaining consistency with the query execute. Let $\epsilon_3 = \mathrm{Adv}^{\mathrm{CCA}}_{\mathrm{Kyber\ KEM}}(\mathbf{A}) + \mathrm{Adv}^{\mathrm{d\text{-}MLWE}}_{R_q^k}(T', n_o)$, where $T' = O(T + (n_o + n_s + n_e)T_{exp})$.

**Claim 4** *For any* **A**, $\mathrm{Adv}^{S2}_{\mathrm{Kyber.PAKE}}(\mathbf{A}) \leq \mathrm{Adv}^{S3}_{\mathrm{Kyber.PAKE}}(\mathbf{A}) + \epsilon_3$

**Proof 4** Let $E3$ be the occurrence of the event Correctpwexec in **S3**. If $E3$ happens, **S2** and **S3** are distinguishable. In Table 3, if Correctpwexec occurs, the event $\mathrm{Testexecpw}(C, i, S, j, pw)$ occurs with two consequences. Given $(A, \alpha, \varphi, ct)$,

- In the query execute, $m = \alpha + (As_1 + e_1)$ and $pk = \varphi + m + \gamma_S$ is set, where $s_1 \leftarrow^r \beta_q^k$ and $e_1 \leftarrow^r \beta_q$. Then, $ct \leftarrow^r D_{ct}$ is chosen.
- Then, **A** makes query $H_{l \in \{2,3\}}(\cdot)$, where $m$ and $pk$ were obtained by query execute. With query $H_1(pw_C)$, $-\gamma_S = As_h + e_h$ is determined, where $s_h \leftarrow^r \beta_q^k$ and $e_h \leftarrow^r \beta_q$. Under these changes, the simulator computes $(ct', K') = \mathrm{Kyber.CCAKEM.Enc}(pk)$. Then, the obtained $(ct', K')$ is added on the possible values's list.

Since the advantage of **A** in Kyber KEM, given in Definition 4, is $\mathrm{Adv}^{\mathrm{CCA}}_{\mathrm{Kyber\ KEM}}(\mathbf{A})$ and the probability of d-MLWE being resolved is $\mathrm{Adv}^{\mathrm{d\text{-}MLWE}}_{R_q^k}(T', n_o)$, Claim 3 is satisfied.

**S4**: Unlike **S3**, **S4** halts when a correct password guess is made against a $\prod_S^j$ or $\prod_C^i$ instance before any query corrupt. In other words, the event Correctpw happens. Then, **A** automatically succeeds.

**Claim 5** *For any* **A**, $\mathrm{Adv}_{\mathrm{Kyber.PAKE}}^{S3}(\mathbf{A}) \leq \mathrm{Adv}_{\mathrm{Kyber.PAKE}}^{S4}(\mathbf{A})$

**Proof 5** If the event Correctpw occurs,

- In an action $\mathrm{CA}_1$ to $\prod_C^i$, if corrupt is not queried after $\mathrm{Testpw}!(C, i, S, pw_C)$, **S4** halts and **A** succeeds.
- In a query $H_{l \in \{2,3\}}(\cdot)$, if corrupt is not queried after $\mathrm{Testpw}^\star(S, j, C, pw_C)$, **S4** halts and **A** succeeds.

Claim 5 is satisfied as these changes will only increase the win probability of **A**.

**S5**: Unlike **S4**, **S5** halts when **A** guesses a password against the partner instances $\prod_S^j$ and $\prod_C^i$. In other words, the event Pairedpwguess happens. Then, **A** fails.

**Claim 6** *For any* **A**, $\mathrm{Adv}_{\mathrm{Kyber.PAKE}}^{S4}(\mathbf{A}) \leq \mathrm{Adv}_{\mathrm{Kyber.PAKE}}^{S5}(\mathbf{A}) + 4n_s \mathrm{Adv}_{R_q^k}^{\mathrm{d-MLWE}}(T', n_o) + \mathrm{Adv}_{\mathrm{Kyber\ KEM}}^{\mathrm{CCA}}(\mathbf{A})$

**Proof 6** For some $\{C, i, S, j\}$, if Pairedpwguess occurs, a $\mathrm{Testpw}(C, i, S, j, pw_C)$ also occurs. In this event, there is a partnership between $\prod_C^i$ and $\prod_S^j$. Let $d \leftarrow^r \{1, 2, \ldots, n_s\}$ be chosen and $(A, \alpha, \varphi, ct)$ is given. In **S5**, Algorithm 2 changes are simulated by **A**.

Since the ROM security of Kyber KEM, given in Definition 4, is $\mathrm{Adv}_{\mathrm{Kyber\ KEM}}^{\mathrm{CCA}}(\mathbf{A})$ and the probability of d-MLWE being solved with send queries is $4n_s \mathrm{Adv}_{R_q^d}^{\mathrm{d-MLWE}}(\mathbf{A})$, Claim 5 is satisfied.

**S6**: Unlike **S5**, in **S6**, there is an internal password oracle that can know all passwords for a given client/server pair and test the correctness of the provided password.

**Claim 7** *For any* **A**, $\mathrm{Adv}_{\mathrm{Kyber.PAKE}}^{S5}(\mathbf{A}) = \mathrm{Adv}_{\mathrm{Kyber.PAKE}}^{S6}(\mathbf{A})$

**Proof 7** Using the password oracle,

- All passwords are generated during initialization and special passwords can be tested in the following way. If $pw = pw_C$, the output of $\mathrm{testpw}(C, pw)$ is True. Otherwise, the output is False.
- All $\mathrm{corrupt}(U)$ is accepted and answered.

In **S6**, $\mathrm{Testpw}(C, i, S, pw)$ for $\prod_C^i$, $\mathrm{Testpw}(S, j, C, pw)$ for $\prod_S^j$, and $\mathrm{Testpw}(C, pw)$ for password oracle queries are checked whether Correctpw occurs. So, **S5** and **S6** can be completely indistinguishable. Claim 6 is satisfied.

In **S6**, **A** has two ways to gain a non-negligible advantage against Kyber.PAKE.

- *Online dictionary attack:* CDF-Zipf model, given in Definition 8, limits the probability of Correctpw event in the proposed Kyber.PAKE since Correctpw event is **A**'s successful obtaining of the password through online dictionary attacks. In other words, $Pr[\mathrm{Correctpw}] = C' \cdot n_{\mathrm{op}}^f + \mathrm{negl}(\kappa)$.

---

**Algorithm 1** S2 Queries and Answers

- In an execute$(C, i, S, j)$ query, $m = As + e$, where $s \leftarrow^r b_\eta^k$ and $e_i \leftarrow^r b_\eta$, $pk \leftarrow^r D_{pk}$, $ct \leftarrow^r D_{ct}$, $\{K, K'''\} \leftarrow^r \{0,1\}^k$, and $\{ssk_2^j = ssk_1^i\} \leftarrow^r \{0,1\}^k$.
- In a CA$_0$ action to $\prod_C^i$, $m = As + e$, where $s \leftarrow^r b_\eta^k$ and $e_i \leftarrow^r b_\eta$.
- In a SA$_1$ action to $\prod_S^j$, $pk \leftarrow^r D_{pk}$, $ct \leftarrow^r D_{ct}$, $K \leftarrow^r \{0,1\}^k$, and $\{K', ssk_2^j\} \leftarrow^r \{0,1\}^k$.
- In a CA$_1$ action to $\prod_C^i$:

    - As a result of this query, if a Testpw!$(C, i, S, pw_C)$ happens, then $K'''$ and $ssk_1^i$ are set to the associated value of Testpw$(C, i, S, pw_C, 2)$ and Testpw$(C, i, S, pw_C, 3)$.
    - If $\prod_C^i$ has a partner $\prod_S^j$, $ssk_2^j = ssk_1^i$. Then, $K''' \leftarrow^r \{0,1\}^k$.
    - If not, $\prod_C^i$ aborts.

- As a result of an SA$_2$ action, if one of the following conditions is satisfied, it terminates. If not, $\prod_S^j$ aborts.

    - If an Testpw!$(S, j, C, pw_C)$ happens, or $\prod_S^j$ has a partner $\prod_C^i$.

- As a result of an $H_{l \in \{2,3\}}(C, S, m, \gamma_S, pk, K)$, if one of the following conditions is met, the output is determined by considering the associated value of the event. If not, the output is randomly chosen from $\{0,1\}^k$.

    - If a Testpw$(S, j, C, pw_C, l)$ or a Testexecpw$(C, i, S, j, pw_C)$ happens.

---

**Algorithm 2** S5 Changes

- For the d-th send$(C, i', S)$ query to $\prod_C^{i'}$, $m = \alpha$ is set.
- In a send$(S, j, < C, m, seed >)$, $pk = \varphi + m + \gamma_S$ is computed.
- In a send$(C, i', < pk, ct, K >)$, if there is no partner for $\prod_C^{i'}$, the output is 0 and **S5** halts.
- Let $\prod_S^j$ and $\prod_C^{i'}$ be partner after its send$(S, j, < C, m, seed >)$ in a send$(S, j, K')$ query to $\prod_S^j$. If the instances have no partnership after this query and Correctpw is not tested, $\prod_S^j$ aborts.
- Then, **A** makes $H_{l \in \{2,3\}}(\cdot)$ query, where $m$ and $pk$ were obtained with $\prod_C^{i'}$. The output of $H_1(pw_C)$ query is defined by $-\gamma_S = As_h + e_h$, where $s_h \leftarrow^r b_\eta^k$ and $e_h \leftarrow^r b_\eta$. Under these changes, the simulator computes $(ct', K') = \text{Kyber.CCAKEM.Enc}(pk)$. Then, the obtained $(ct', K')$ is added to the possible values list.

---

- *A test query:* Let $\prod_U^i$ be a fresh instance. Then, **A** makes a query test$(U, i)$ to $\prod_U^i$. Since the view of **A** is completely independent of $ssk_U^i$, $\Pr[\text{Suc}_{\text{Kyber.PAKE}}^{S6}(\mathbf{A}) | \neg \text{Correctpw}] = 1/2$.

    By considering these two options, Eq. (6) is obtained.

$$
\Pr[\text{Suc}_{\text{Kyber.PAKE}}^{S6}(\mathbf{A})] \leq \overbrace{\Pr[\text{Correctpw}]}^{C' \cdot n_{\text{op}}^f} + \overbrace{\Pr[\text{Suc}_{\text{Kyber.PAKE}}^{S6}(\mathbf{A}) | \neg \text{Correctpw}]}^{1/2} \overbrace{Pr[\neg \text{Correctpw}]}^{1 - C' \cdot n_{\text{op}}^f}
$$

$$
\leq 1/2(1 + C' \cdot n_{\text{op}}^f) \tag{6}
$$

**Table 4 Parameter set.**

| Scheme | Security level | k | n | q | $\eta$ | $\eta_1$ | $\eta_2$ | $(d_u, d_v)$ | $\delta$ |
|---|---|---|---|---|---|---|---|---|---|
| MLWE.PAKE (*Ren, Gu & Wang, 2023*) | 116 | 2 | 256 | 7,681 | 13 | x | x | x | $2^{-53.4}$ |
| | 177 | 3 | 256 | 7,681 | 8 | x | x | x | $2^{-97.4}$ |
| | 239 | 4 | 256 | 7,681 | 6 | x | x | x | $2^{-131.6}$ |
| Proposed Kyber.PAKE | 128 | 2 | 256 | 3,329 | x | 3 | 2 | (10,4) | $2^{-131}$ |
| | 192 | 3 | 256 | 3,329 | x | 2 | 2 | (10,4) | $2^{-164}$ |
| | 256 | 4 | 256 | 3,329 | x | 2 | 2 | (11,5) | $2^{-174}$ |

According to Eq. (2), $\mathrm{Adv}^{S6}_{\mathrm{PAKE}}(\mathbf{A}) = 2\mathrm{Pr}[\mathrm{Suc}^{S6}_{\mathrm{Kyber.PAKE}}(\mathbf{A})] - 1 \leq C' \cdot n^f_{\mathrm{op}}$. If Eq. (2) is rewritten by considering Claims (2)–(7), Eq. (7) is obtained.

$$\mathrm{Adv}^{\mathbf{S}}_{\mathrm{Kyber.PAKE}}(\mathbf{A}) \leq 2\left|Pr[\mathrm{Suc}^{S0}_{\mathrm{Kyber.PAKE}}] - \frac{1}{2}\right| = 2\left|Pr[\mathrm{Adv}^{S0}_{\mathrm{Kyber.PAKE}}] - Pr[\mathrm{Adv}^{S6}_{\mathrm{Kyber.PAKE}}]\right|$$

$$= 2\left( \overbrace{\left|Pr[\mathrm{Adv}^{S0}_{\mathrm{Kyber.PAKE}}] - Pr[\mathrm{Adv}^{S1}_{\mathrm{Kyber.PAKE}}]\right|}^{\leq \frac{(n_e+n_s)(n_e+n_s+n_o)}{q^{nk}}} + \overbrace{\left|Pr[\mathrm{Adv}^{S1}_{\mathrm{Kyber.PAKE}}] - Pr[\mathrm{Adv}^{S2}_{\mathrm{Kyber.PAKE}}]\right|}^{\leq \frac{n_o}{q^{nk}} + \frac{n_s}{2^\kappa}} \right.$$

$$+ \overbrace{\left|Pr[\mathrm{Adv}^{S2}_{\mathrm{Kyber.PAKE}}] - Pr[\mathrm{Adv}^{S3=S4}_{\mathrm{Kyber.PAKE}}]\right|}^{\mathrm{Adv}^{\mathrm{CCA}}_{\mathrm{Kyber\ KEM}}(\mathbf{A}) + \mathrm{Adv}^{\mathrm{d\text{-}MLWE}}_{R^k_q}(\mathbf{A})} + \overbrace{\left|Pr[\mathrm{Adv}^{S4}_{\mathrm{Kyber.PAKE}}] - Pr[\mathrm{Adv}^{S5}_{\mathrm{Kyber.PAKE}}]\right|}^{4n_s\mathrm{Adv}^{\mathrm{d\text{-}MLWE}}_{R^k_q}(\mathbf{A}) + \mathrm{Adv}^{\mathrm{CCA}}_{\mathrm{Kyber\ KEM}}(\mathbf{A})} \qquad (7)$$

$$\left. + \overbrace{\left|Pr[\mathrm{Adv}^{S5}_{\mathrm{Kyber.PAKE}}] - Pr[\mathrm{Adv}^{S6}_{\mathrm{Kyber.PAKE}}]\right|}^{1/2(1+C' \cdot n^f_{\mathrm{op}})} \right)$$

Since $\mathrm{Adv}^{\mathbf{S}}_{\mathrm{Kyber.PAKE}}(\mathbf{A}) \leq C' \cdot n^f_{\mathrm{op}} + O\left(\frac{(n_e+n_s)(n_e+n_s+n_o)+n_o}{q^{nk}} + \frac{n_s}{2^\kappa} + \mathrm{Adv}^{\mathrm{CCA}}_{\mathrm{Kyber\ KEM}}(\mathbf{A}) + n_s\mathrm{Adv}^{\mathrm{d\text{-}MLWE}}_{R^k_q}(\mathbf{A})\right)$, Theorem 3 is hold.

# REFERENCE IMPLEMENTATION AND COMPARISON RESULTS

In this section, the reference implementation of Kyber.PAKE is presented in terms of cost, CPU cycle, running time, and memory usage. In addition, detailed comparisons with literature proposals based on performance evaluations are also provided.

The implementation of Kyber.PAKE is written in C (*Dursun, 2023a*) based on Kyber KEM's reference C codes and PAK design components. The performance results are obtained by using a computer with a 2.5 GHz dual-core Intel Core i5 processor and 8 GB RAM. The obtained performance evaluation is compared with MLWE.PAKE scheme (*Ren, Gu & Wang, 2023*) since it is the only MLWE-based PAKE in the literature. For these two schemes, the parameter sets are recalled in Table 4.

To obtain comparisons in terms of running time, MLWE.PAKE and our implementation are run 1,000 times. Based on the main processes or functions, the CPU cycles are determined for 128-bit security level and presented in Table 5. It can be seen from Table 5, the proposed Kyber.PAKE scheme needs fewer average and media CPU cycles due to the small size of the parameter set and its efficient/simple structure components.

**Table 5** CPU cycle comparision for 128-bit security level.

| Functions/Processes | MLWE.PAKE (*Ren, Gu & Wang, 2023*) | | Kyber.PAKE | |
|---|---|---|---|---|
| | *Avg.* | *Med.* | *Avg.* | *Med.* |
| GenMatrix() | 31,108 | 27,997 | **24,188** | **22,109** |
| PolyGetNoise() | 4,412 | 4,112 | **3,943** | **3,512** |
| PolyNtt() | 13,429 | 12,664 | **7,798** | **7,443** |
| PolyvecNtt() | 33,170 | 27,061 | **15,024** | **14,121** |
| PolyvecInvntt() | 30,621 | 26,460 | **21,248** | **19,906** |
| OkcnCon() | 17,699 | 16,058 | x | x |
| OkcnRec() | 3,489 | 3,297 | x | x |
| Kyber.CCAKEM.Enc() | x | x | 182,018 | 165,958 |
| Kyber.CCAKEM.Dec() | x | x | 193,497 | 173,239 |
| $C_0$ | 195,201 | 173,157 | **143 497** | **124 864** |
| $S_0$ | 307,547 | 265,276 | **224,537** | **183,024** |
| $C_1$ | 133,436 | 117,676 | 256,217 | 228,652 |
| $S_1$ | 40,446 | 30,603 | 59,907 | 57,807 |

**Notes.**
Bold values indicate cases where the proposed scheme provides better results than the compared ones in terms of the analyzed metrics.

Table 6 gives the average run time results, which is constructed by considering common components, scheme phases, hash functions, and reconciliation structures. Due to its parameter set, Kyber.PAKE provides better results in generating pk ($A$) with GenMatrix() and hash functions. Since KEM structures such as encapsulation and decapsulation, which have additional components for security, are used in Kyber.PAKE, it requires more runtime than MLWE.PAKE in terms of reconciliation. Considering the total times on the client and server sides, MLWE.PAKE is better on the client side. One of the reasons is that in MLWE.PAKE, key generation takes place on both the client and server sides, while it is only made on the client side of Kyber.PAKE. Different design approaches, reconciliation functions, and parameter sets also affect.

The computational cost evaluation of lattice-based two-party PAKEs that were constructed by following the one-phase idea is also provided with Table 7. Even if the selected schemes were designed under the same approach, the main securities were captured with different hard problems. So, message size-based evaluation is just presented in Table 7.

In Table 7, the provided results are obtained in the following way. It can be seen in Kyber.PAKE's protocol flow, $\{\text{seed}, \text{cid}, m_{\text{bytes}}, K'''\}$ are transferred to the server. On the server side, $\{pk, ct, K\}$ components are sent to the client. According to the selection or computations of these values, it is known that $\{\text{seed}, \text{cid}, K, K'''\}$ are fixed 32-byte and $\{m_{\text{bytes}}, pk_{\text{bytes}}\} = k \cdot 384$, where $k$ is determined differently for each security levels.

Let's show how the message sizes of Kyber.PAKE is computed for 128−bit security level.

- Client to Server: $\text{seed} + \text{cid} + m_{\text{bytes}} + K''' = 32 + 32 + (2 \cdot 384) + 32 = 864$ bytes.
- Server to Client: $pk_{\text{bytes}} + ct_{\text{bytes}} + K = (2 \cdot 384) + 768 + 32 = 1{,}568$ bytes.

**Table 6  Running times in microseconds.**

| Scheme security level | (*Ren, Gu & Wang, 2023*) 116 | Kyber.PAKE 128 | (*Ren, Gu & Wang, 2023*) 177 | Kyber.PAKE 192 | (*Ren, Gu & Wang, 2023*) 239 | Kyber.PAKE 256 |
|---|---|---|---|---|---|---|
| GenMatrix() | 13.893 | 9.256 | 27.504 | 21.648 | 49.979 | 38.713 |
| OkcnCon() | 7.058 | x | 5.920 | x | 5.293 | x |
| OkcnRec() | 1.425 | x | 1.622 | x | 1.655 | x |
| Kyber.CCAKEM.Enc() | x | 69.133 | x | 110.894 | x | 152.360 |
| Kyber.CCAKEM.Dec() | x | 72.362 | x | 117.631 | x | 177.787 |
| shake128 | 2.656 | 2.390 | 2.422 | 2.923 | 3.036 | 2.397 |
| shake256 | 13.386 | 11.328 | 16.680 | 16.235 | 22.904 | 21.586 |
| $C_0$ | 87.456 | 52.449 | 112.925 | 88.894 | 155.515 | 141.205 |
| $S_0$ | 126.205 | 71.135 | 155.530 | 114.015 | 202.895 | 165.042 |
| $C_1$ | 50.409 | 93.443 | 70.565 | 150.637 | 90.342 | 217.362 |
| $S_1$ | 12.942 | 21.781 | 16.689 | 32.918 | 21.930 | 42.184 |
| *Total client* | 138.865 | 145.892 | 183.490 | 239.531 | 245.857 | 358.567 |
| *Total server* | 139.147 | **92.916** | 172.219 | **146.993** | 224.825 | **207.256** |

**Notes.**
Bold values indicate cases where the proposed scheme provides better results than the compared ones in terms of the analyzed metrics.

**Table 7  A comparison for message sizes of lattice-based PAK PAKE schemes.**

| Reference | Hardness | Security level | C | S | C+S |
|---|---|---|---|---|---|
| *Gao et al. (2017)* | RLWE | 82 | 3,904 | 4,000 | *7,904* |
| *Ding et al. (2017)* | RLWE | 76 | 4,136 | 4,256 | *8,392* |
| *Yang et al. (2019)* | RLWE | 206 | 1,864 | 2,592 | *4,456* |
| *Ren, Gu & Wang (2023)* | MLWE | 116 | 928 | 1,056 | *1,984* |
| | | 177 | 1,344 | 1,472 | *2,816* |
| | | 239 | 1,760 | 1,888 | *3,648* |
| Kyber.PAKE | MLWE | 128 | **864** | 1,568 | *2,432* |
| | | 192 | **1,248** | 2,272 | *3,520* |
| | | 256 | **1,632** | 3,136 | *4,768* |

**Notes.**
Bold values indicate cases where the proposed scheme provides better results than the compared ones in terms of the analyzed metrics.

**Remark 2**  *The comparisons in* Tables 5 *and* 6 *are conducted by assuming that (Ren, Gu & Wang, 2023) presents approximately the same security levels. Note that Kyber.PAKE will provide better results when the parameters are changed to achieve the same security levels.*

Using the Kyber.PAKE C codes (*Dursun, 2023a*), Java codes (*Dursun, 2023b*) are also written to demonstrate the usability of the proposed scheme on mobile devices. In the implementation, a computer with a 2.5 GHz dual-core Intel Core i5 processor and 8 GB RAM is used as the server. Samsung Galaxy A51 (8 Cores) with 4x 2.3 GHz ARM Cortex-A73 main processor and 4x 1.7 GHz ARM Cortex-A53 co-processor with 2.3 GHz CPU frequency device is utilized as the client. Kyber.PAKE mobile results in terms of

**Table 8** Implementation results of Kyber.PAKE on mobile device.

| Security level | Phase | Running time[*] | Memory usage | CPU usage |
|---|---|---|---|---|
| | $C_0$ | 745.918 | 104.2 KB | %8 |
| | $S_0$ | 880.761 | 88.6 KB | %10 |
| **128** | $C_1$ | 997.569 | 168.3 KB | %10 |
| | $S_1$ | 446.311 | 0.4 KB | %7 |
| | Total client | 1743.487 | 272.5 KB | %18 |
| | Total server | 1327.072 | 89 KB | %17 |
| | $C_0$ | 918.225 | 148.2 KB | %10 |
| | $S_0$ | 945.361 | 133.7 KB | %11 |
| **192** | $C_1$ | 1215.136 | 211.4 KB | %12 |
| | $S_1$ | 611.217 | 0.4 KB | %8 |
| | Total client | 2133.361 | 359.6 KB | %22 |
| | Total server | 1556.578 | 134.1KB | %19 |
| | $C_0$ | 1211.843 | 177.8 KB | %11 |
| | $S_0$ | 1388.745 | 171.1 KB | %13 |
| **256** | $C_1$ | 1811.257 | 297.2 KB | %14 |
| | $S_1$ | 874.413 | 0.5 KB | %10 |
| | Total client | 3023.1 | 475 KB | %25 |
| | Total server | 2236.158 | 171.6 KB | %23 |

**Notes.**
[*]In microseconds.
Bold values indicate cases where the proposed scheme provides better results than the compared ones in terms of the analyzed metrics.

runtime, memory, and CPU usage are given in Table 8, which is obtained by running all the phases of the client and server 1,000 times.

The mobile device compatibility of Kyber.PAKE is also analyzed regarding energy, memory, and CPU usage. For 128-bit security, each sub-processes of Kyber.PAKE is examined with the Android Profiler tool of Android Studio and given in Fig. 2. As a case scenario, the energy consumption metric is also detailed in Fig. 3.

Figures 2 and 3 show that although the proposed PAKE does not contain any optimization or improvement techniques, it has relatively low resource usage. So, we can say that constructed Kyber.PAKE will be preferred to obtain the post-quantum secure mobile environment.

**Remark 3** *Note that two other lattice-based PAKE schemes (Dabra, Bala & Kumari, 2020; Ding, Cheng & Qin, 2022; Seyhan & Akleylek, 2024) for two-party mobile device security were proposed using different approaches, hardness, and additional properties. When we checked the proposals, no source code was given, and the results were not provided for all metrics, such as memory, CPU, and energy usage. Therefore, we compared MLWE-based PAKEs in terms of running times and presented a computational cost examination for all two-party PAK PAKEs.*

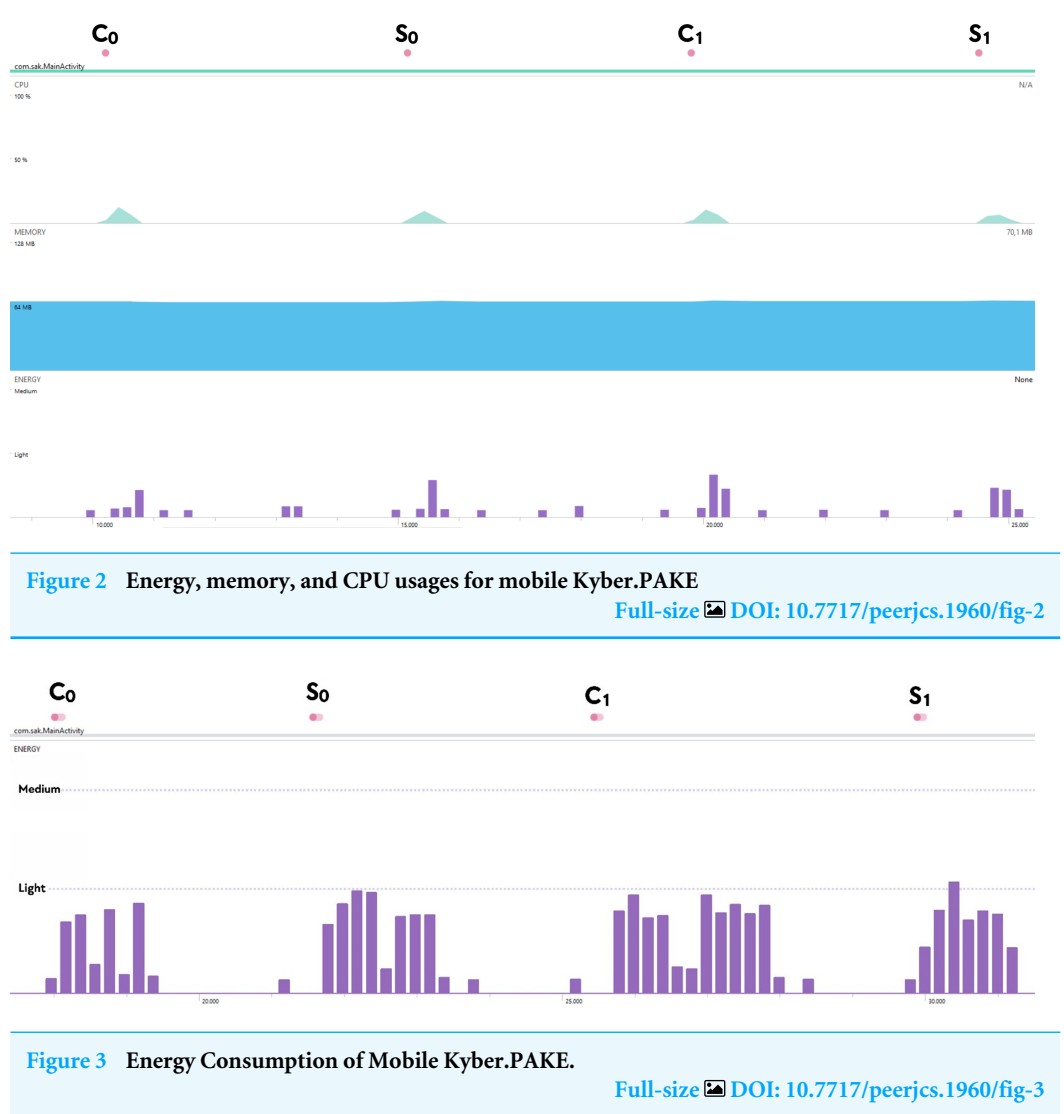

**Figure 2** Energy, memory, and CPU usages for mobile Kyber.PAKE

**Figure 3** Energy Consumption of Mobile Kyber.PAKE.

# CONCLUSION AND FUTURE DIRECTIONS

In this article, a two-party PAKE version of Kyber KEM is constructed to provide a proposal for post-quantum PAKE requirements by adapting the standard algorithms for different purposes and usage areas. Kyber.PAKE is obtained by adjusting the traditional PAK design idea to the MLWE problem and Kyber KEM functions. In the password-authenticated shared key generation, it is shown that explicit authentication and PFS properties are captured. The security of Kyber.PAKE is analyzed by considering dictionary attack resistance under the ROM assumptions. In these examinations, the CDF-Zipf model is also added to determine more realistic security proofs by considering the real-world distribution of the passwords. The reference implementation results show that the Kyber.PAKE scheme can be one of the best choices in post-quantum era security in terms of run-time, memory, and CPU usage. The mobile device usage of the proposed PAKE is also analyzed by providing reference Java implementation. As far as we know, the

constructed Kyber.PAKE is the first PAKE adaptation of the NIST PQC KEM standard with mobile environment compatibility. As a future direction, the security examination of Kyber.PAKE will be extended by defining quantum random oracle model assumptions and the resource-limited device usage will be provided by making arithmetic optimizations and improvements.

### Funding
This work was supported by the Estonian Research Council Grant no. PRG946 and TUBITAK under grant no. 121R006. The funders had no role in study design, data collection and analysis, decision to publish, or preparation of the manuscript.

### Grant Disclosures
The following grant information was disclosed by the authors:
The Estonian Research Council: PRG946.
TUBITAK: 121R006.

### Competing Interests
Sedat Akleylek is the Section Editor for Cryptography.

### Author Contributions
- Kübra Seyhan conceived and designed the experiments, performed the experiments, analyzed the data, performed the computation work, prepared figures and/or tables, authored or reviewed drafts of the article, and approved the final draft.
- Sedat Akleylek conceived and designed the experiments, performed the experiments, analyzed the data, performed the computation work, prepared figures and/or tables, authored or reviewed drafts of the article, and approved the final draft.
- Ahmet Faruk Dursun conceived and designed the experiments, performed the experiments, analyzed the data, prepared figures and/or tables, authored or reviewed drafts of the article, and approved the final draft.

### Data Availability
The source codes are available in the Supplemental Files, GitHub and Zenodo:
- https://github.com/afDursun/Kyber-PAKE-Mobile
- afDursun. (2024). afDursun/Kyber-PAKE-Mobile: kyber-pake-mobile (1.0.0). Zenodo. https://doi.org/10.5281/zenodo.10784698
- https://github.com/afDursun/Kyber-PAKE-C
- afDursun. (2024). afDursun/Kyber-PAKE-C: kyber-pake-c (1.0.0). Zenodo. https://doi.org/10.5281/zenodo.10784450.

### Supplemental Information
Supplemental information for this article can be found online at http://dx.doi.org/10.7717/peerj-cs.1960#supplemental-information.

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
