# Peer review of "Password authenticated key exchange-based on Kyber for mobile devices"

_PeerJ Computer Science, doi:10.7717/peerj-cs.1960_

## Round 0.1 · original submission · Minor Revisions

The authors must revise according to the reviewers. In the case of Reviewer 1, the references recommended should be added only if they add value to the paper.

**Language Note:** The review process has identified that the English language must be improved. PeerJ can provide language editing services - please contact us at copyediting@peerj.com for pricing (be sure to provide your manuscript number and title). Alternatively, you should make your own arrangements to improve the language quality and provide details in your response letter. – PeerJ Staff

·

Basic reporting

The work has a good technical definition and specifications.

I believe it will be able to contribute to the magazine and readers with some improvements. I suggest that the authors clarify the open problems the work aims to solve in the introduction.

A comparative section with the state of the art about the work developed is necessary. Include works published in the last 5 years.

These works can contribute to the work developed:

https://doi.org/10.3390/s19194312

https://doi.org/10.3390/a16010038

Experimental design

In section 5, it is unclear power consumption, battery usage and definitions of data type, quantity and size for all tests performed.

Validity of the findings

The conclusions are poor, I strongly suggest that the authors value the effort and clearly describe the problems solved, the concrete results obtained and compared, the news and especially the scientific contributions of the article. Finally, I strongly suggest you add any open points for readers to develop in future work.

Additional comments

I hope I have contributed to improving the work.

Cite this review as

·

Basic reporting

This work is interesting. It concerns a proposal for a new two-party PAKE version of Kyber KEM. The work is done professionally. Formal results prove the content.

Experimental design

The experiment is described well. However, the work is not readable. Some of the details are not explained well. The work is not self-contained, i.e., it could not be read without referenced papers.

Validity of the findings

The validity of the findings is proven.

Additional comments

I propose adding more comments and explanations to the work as well as editing it carefully.

Reviewer 3 ·

Basic reporting

The paper presents a novel password-authenticated key exchange (PAKE) scheme based on Kyber, tailored for mobile environments. Below are some areas of improvement and suggestions to enhance the clarity, coherence, and depth of the manuscript.

The introduction is clear and states the work motivation clearly;

Some acronyms are not described in the text. As an example: MLWE, CDF-Zip, and others;

The paper is written in English at a proficient level, and it is well-structured. However, the article is reasonably complex, so understanding that it is a complete subject, it would be appropriate to add some explanations and/or concepts so that more readers can understand and replicate the work in its totality. Some references may help.

Experimental design

I don’t have concerns in this matter. Once more, the experiments are well-designed and described, but the complexity inherent to the subject is not negligible. Some references may help to enlighten a broader audience.

Validity of the findings

No concerns were, as well.

Cite this review as

·

Basic reporting

The article is written in English and uses clear, concise, technically correct text. The article complies with professional standards of courtesy and expression.
Literature references, adequate field history/context provided.
Professional article structure, figures, tables are appropriate. Data has been shared.
In the study, all results related to the hypothesis are given.
The results of the study were tried to be expressed clearly with tables.

Experimental design

This is a study within the Scope of the Journal.
In the first part, the research question could have been defined more clearly. The research question could be supported with references.
The problem proposed to be solved in the study was carried out at an appropriate technical standard and the results were supported by tables and figures.
In the study, the method and method are explained in the 2nd and 3rd section.

Validity of the findings

In the study, the data used for the Results are acceptable and ready for use.
The results are expressed appropriately. The data in the tables and figures have been interpreted correctly. In the study, technical and mathematical aspects for the solution of the mentioned problem are appropriately stated in the 3rd and 4th chapters.

Additional comments

In the study, the data used for the Results are acceptable and ready for use.
The results are expressed appropriately. The data in the tables and figures have been interpreted correctly. In the study, technical and mathematical aspects for the solution of the mentioned problem are appropriately stated in the 3rd and 4th chapters.

In the conclusion part, suggestions can be made about future studies.

Cite this review as

---

## Round 0.2 · accepted · Accept

The authors considered the minor issues, and it can be accepted.